# Towards General Single-Utensil Food Acquisition with Human-Informed Actions

**Ethan K. Gordon**[*]
University of Washington
`ekgordon@cs.uw.edu`

**Amal Nanavati**[*]
University of Washington
`amaln@cs.uw.edu`

**Ramya Challa**
Oregon State University

**Bernie Hao Zhu**
University of Washington

**Taylor A. Kessler Faulkner**
University of Washington

**Siddhartha S. Srinivasa**
University of Washington

**Abstract:** Food acquisition with common general-purpose utensils is a necessary component of robot applications like in-home assistive feeding. Learning acquisition policies in this space is difficult in part because any model will need to contend with extensive state and actions spaces. Food is extremely diverse and generally difficult to simulate, and acquisition actions like skewers, scoops, wiggles, and twirls can be parameterized in myriad ways. However, food's visual diversity can belie a degree of physical homogeneity, and many foods allow flexibility in how they are acquired. Due to these facts, our key insight is that a small subset of actions is sufficient to acquire a wide variety of food items. In this work, we present a methodology for identifying such a subset from limited human trajectory data. We first develop an over-parameterized action space of robot acquisition trajectories that capture the variety of human food acquisition technique. By mapping human trajectories into this space and clustering, we construct a discrete set of 11 actions. We demonstrate that this set is capable of acquiring a variety of food items with $\geq 80\%$ success rate, a rate that users have said is sufficient for in-home robot-assisted feeding. Furthermore, since this set is so small, we also show that we can use online learning to determine a sufficiently optimal action for a previously-unseen food item over the course of a single meal.

## 1 Introduction

Eating is a fundamental part of the human experience, and robots can play an important role in facilitating the transfer of food from farm to kitchen to plate to mouth. But the prerequisite process of actually picking food up can be tricky. Food is fragile and can fall apart with excessive forces or stick to both the robot and the environment. Food is visually diverse and hard to simulate. In industrial transport and packaging settings, specialized end-effectors can utilize suction [1] or soft enveloping links [2] to safely and reliably grasp a variety of foods. However, for in-home food manipulation, such hardware can be unavailable, impractical due to limited space, or uncomfortable for humans to interact with. In particular, this work is motivated by the application of robot-assisted feeding for those who cannot eat on their own (1.8 million people in the United States alone [3]). In this setting, there is strong coupling between the problems of food acquisition and mouth transfer [4]. Therefore, we focus on the problem of acquiring food with common, single-piece utensils (i.e. a fork or spoon) with which users are already familiar.

Formally, our goal is to learn a map from food context (e.g., RGBD images of the food item) to a sufficiently good utensil trajectory and control strategy to acquire the food. In the assisted feeding context, previous work [5] suggests that that "sufficiently good" is on the order of an $80\%$ success rate, depending on the level of mobility impairment.

7th Conference on Robot Learning (CoRL 2023), Atlanta, USA.

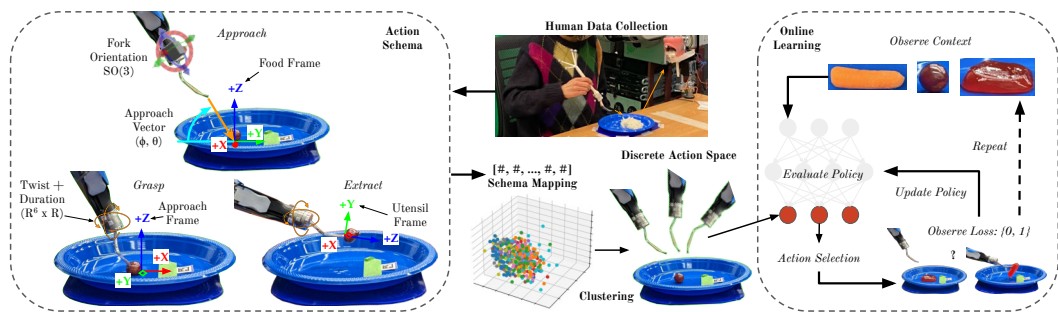

Figure 1: *(Left)*: Visual description of the action schema. Robot motions are in orange. Reference frames are represented as three-color axes with X in red, Y in green, and Z in blue. *(Right)*: General food acquisition pipeline. Human data is collected, mapped into the schema, and clustered into a discrete action space. This space is small enough to treat food acquisition as a contextual bandit to learn online the optimal action for new food items.

Humans are generally experts at food acquisition, so a natural approach to this problem is to use a human dataset collected on a variety of food items to learn this map directly. This approach runs into sample complexity issues, as food is hard to simulate, and it is difficult to capture the quantity of real world data necessary to cover the diversity of food and the diversity of possible ways to acquire it. Previous work [6, 7] got around this issue by using human data as a qualitative starting point for the manual design of a small set of actions with some success on a limited set of food items.

Supported by this work and other work exploiting the haptic similarities between food items [8], our key insight is that a very small subset of the space of possible acquisition actions is sufficient to acquire almost all food items that a human is capable of picking up with a fork. We can capture this subset in a principled, data-driven way by observing a relatively small number of human acquisition trajectories on arbitrary food. By parameterizing these trajectories within an interpretable metric space, we can use off-the-shelf clustering to create an even smaller representative discrete space. We empirically show that this discrete space is simultaneously large enough to effectively cover the space of food contexts and small enough that online learning can identify the best action for multiple new food contexts over the course of a 30 minute meal.

In summary, this work presents three contributions in the field of robotic food acquisition. (1) In Section 3 we present the schema that defines an over-parameterized action space to capture human food acquisition techniques with a common single-piece utensil (i.e. fork or spoon). (2) In Section 4, we present a dataset of human food acquisition trajectories on a user-motivated variety of food items. (3) In Section 4.2, we describe both a method for distilling discrete actions from human data and a set of 11 actions constructed from our dataset. This method is tailored to in-home robot-assisted feeding, but we believe a similar structure could be used for other applications where the key insight holds.

In Section 5 we demonstrate that the discrete action set is sufficient to acquire a variety of food items that are visually dissimilar from those used in the human study. We demonstrate in Section 6 that we can quickly use off-the-shelf online learning techniques to determine a sufficiently optimal action for previously-unseen food items within 13 trials per item. Finally, Section 7 discusses avenues for future work in both general food manipulation research and the application of robot-assisted feeding.

## 2 Related Work

### 2.1 Robot-Assisted Feeding: Food Manipulation

Robot-assisted feeding has been explored in industry and academia, yet still contains many unsolved problems. Commercial table-mounted systems [9, 10, 11, 12, 13, 14, 15, 16] are available, but work with fixed trajectories for food acquisition. This can result in issues such as food being pushed off of

the plate, as found in a study of the Bestic system [17]. There are also teleoperated options available, such as [18], but users can have difficulties using a fully teleoperated system [19].

Food acquisition is a fundamental part of robotic feeding devices [20]. Some prior work creates specialized tools for food manipulation [1, 21, 22, 2]. We focus instead on food acquisition with a fork, as it is a common household utensil with which users are familiar. Other research that has focused on picking up food with forks uses a limited set of skewering actions, which works well for a small set of food items but has difficulties on more varied items [23, 24, 7]. Other work added variety with utensil swapping [25], which adds hardware complexity, but kept the set of food items and action trajectories small.

Further work in food acquisition uses vision or haptic data to improve the choice of acquisition action. Vision can be used to classify visually different food items, and haptic data can assist in identifying foods that look different but require similar actions (e.g. grapes and cherry tomatoes). Some prior works require additional "probing" actions for every food item [26, 27, 6] or specialized sensors beyond force torque sensors [28]. Other work uses haptic data to improve food acquisition during the feeding process, but the expert-designed action space does not cover the variety of food necessary for in-home applications [8, 29]. This paper leverages the learning approaches of past work with a human-informed action space that is likely to cover a wider variety of food items that users might want to eat.

## 2.2 Learning Grasps from Human Demonstrations

As humans are generally expert tool users, a plethora of work has gone into transferring those skills to robots [30, 31, 32]. Some work focuses on higher-level task planning [33, 34]. Others learn more granular motions by generating dynamic motion primitives that a model learns to stitch together [35, 36]. Still others investigate a hand-design restricted action space for use with end-to-end models [37]. In contrast, this work looks at leveraging application-specific structure and human data to systematically restrict the action space prior to learning a model.

Other works [38, 39, 40, 41, 42] utilize simulations or extensive human and environment data to learn offline RL policies that yield good generalization performance at test time. In contrast, this work exploits application-specific structure and the simpler contextual-bandit setting to learn a simpler model that can be refined online without the need for simulation or large datasets.

## 3 Acquisition Action Schema

Previous work [6] qualitatively captured a taxonomy of human food acquisition techniques with a fork. This taxonomy included skewering and scooping and highlighted the importance of the approach angle and in-food manipulation strategies (e.g., wiggling for greater pressure or rotation for greater contact area). Yet, that work does not provide a quantitative way to represent those motions so a robot can execute them.

Filling this gap, the following schema describes an acquisition action space that is narrow enough to distill these taxonomic elements but flexible enough to capture variants of those elements (e.g., additional wiggling, or a different approach angle). The action is defined by 26 continuous parameters divided into three phases: approach, grasp, and extraction (Fig. 1, left).

### 3.1 Approach (Pre-Grasp)

This phase captures the fork tilt and and approach angle elements of the qualitative grasp taxonomy.

**Frame Definitions**: Define the world frame with an arbitrary origin and orientation such that -Z is the direction of gravity. We assume the existence of a food manipulation target bounded by an ellipsoid which may possibly intersect a flat plane defined by a table/plate/other surface parallel to the X-Y plane (from here just referred to as the plate). The projection of this ellipsoid onto the plate is the *bounding ellipse* of the food. The *food frame* is the default reference frame in which all parameters

are defined unless otherwise specified. The origin is defined as the center of the bounding ellipse. +Z is aligned with that of the world frame and the X-axis is aligned with the major axis of the bounding ellipse.

**Parameters**: The approach consists of the following 9 parameters: Fork orientation ($SO(3)$), Approach polar and azimuthal angle ($[0, \frac{\pi}{2}] \times [0, 2\pi)$), Target approach point within the food ($\mathbb{R}^3$), Force threshold ($+\mathbb{R}$).

**Implementation**: During implementation, the utensil begins an arbitrary distance from the food and moves in a straight line towards the target approach point until either that point is reached or the force on the utensil exceeds the threshold. Note that the definition of the food frame introduces a $\pi$-rotation symmetry depending on which direction along the X-axis is +X. In this work, this symmetry is broken during the on-robot experiments based on which approach direction is within the robot's workspace and easiest for the on-board planning algorithm.

### 3.2 Grasp

This phase captures the wiggling, twirling, and in-food scooping motions of the qualitative taxonomy.

**Frame Definitions**: Define the *approach frame* as the food frame rotated by the azimuthal angle of the approach direction. Define the *utensil frame* with an origin at the very tip of of the utensil (e.g., between the middle two tines on a fork) such that +Z points along the handle of the utensil and the X-axis goes across the face of the utensil. In other words, the Euler angles in this frame correspond with roll (Z), pitch (X), and yaw (Y). Using this frame instead of the food frame allows the approach and grasp to be parameterized independently. For example, approaching from the side instead of the front should not result in a grasp rotation yawing instead of pitching the fork.

**Parameters**: The grasp consists of the following 9 parameters: Angular velocity in utensil frame ($\mathbb{R}^3$), Linear velocity in approach frame ($\mathbb{R}^3$), Duration ($+\mathbb{R}$), Force and torque thresholds ($+\mathbb{R} \times +\mathbb{R}$).

**Implementation**: During implementation, the utensil will execute the provided velocities for the provided duration, or cut short if the force or torque thresholds are reached.

### 3.3 Extraction

This phase captures any stabilizing rotations that take place after the food is on the fork.

**Parameters**: Similarly to grasp, the extraction consists of the following 7 parameters: Angular velocity in utensil frame ($\mathbb{R}^2 \times +\mathbb{R}$), Linear velocity in approach frame ($\mathbb{R}^3$), Duration ($+\mathbb{R}$). While a force and torque threshold can be introduced, it is rendered unnecessary in this work by requiring the extraction motion to move against gravity away from the plate.

**Implementation**: Extraction is implemented the same way "grasp" is.

## 4 Human Bite Acquisition Strategies

Although the 26 dimensional action schema is large, we hypothesized that only specific points (acquisition actions) within this schema will actually be commonly used to acquire food items. To identify those points, we had able-bodied participants acquire a variety of food items and feed them to an actuated mouth. The food items were chosen based on the meals an end-user with C1 quadripledia[1] eats in a week, and included bready items like bagels and pizza, heterogenous items like sandwiches rice and beans, gelatenous items like jello, and stringy items like noodles. As the users acquired the food, a motion capture system captured fork motion, a force-torque sensor on the fork captured haptic factors, and an RGB-D camera above the plate captured visual aspects of the food manipulation (Fig. 2a). More details about the data collection can be found in Appendix A.

---

[1]C1 quadriplegia refers to paralysis of all four limbs as a result of an injury to the first, or top-most, cervical vertebrate.

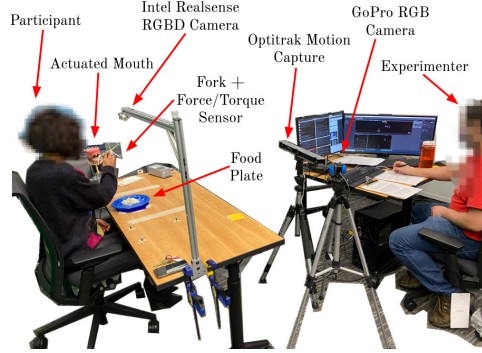
(a) Human food acquisition data collection setup.

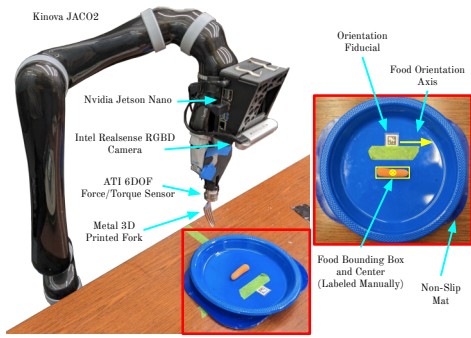
(b) On-robot experiment setup.

Figure 2: Food acquisition trials. For each trial, a single food item was acquired. Food perception (both center-of-mass and orientation) was performed with classical computer vision through a fiducial and color-based background rejection.

## 4.1 Dataset

We published the data gathered here [43] to facilitate future research in food acquisition. This dataset consists of 496 bite acqusion trials, totaling over 1.25 hours of food acquisition data across 9 participants and 13 unique food items.

## 4.2 Human Data Analysis

### 4.2.1 Extracting an Action Schema Point

For each bite's acquisition data, we extracted a point within the acquisition action schema that was close to that motion. We developed the procedure by iteratively extracting action schema points from bites, and then visualizing the actual participant's motion superimposed with the extracted action on a random subset of bites to determine how to improve the extraction procedure. Complete implementation details are covered in Appendix B and our code repository[2].

### 4.2.2 Clustering Actions

We ran k-medoids on the entire dataset of $407$ extracted actions, where each action has 26 dimensions. Notably, the clustering did not consider aspects of the action that would be outputted by the perception system, such as food reference frame, and only included the aspects of the acquisition motion that might generalize across food items. This resulted in $k = 11$ actions (corresponding to the within-cluster-sum-of-square-distances elbow point). A full quantitative parameterization and video of each action are provided in the supplementary materials. Qualitatively, we observed emergent behavior consistent with findings in previous work [6], such as in-food wiggling, tilted extraction, and the use of vertical tines for high force.

### 4.2.3 From Human to Robot Actions

Although this procedure outputs representative actions for the motions that participants took when acquiring food items, it also learns some aspects of motion that are particular to the morphology of a human arm. For example, participants' motions tended to approach food from the right, since they were right-handed and feeding a person to their left. However, since the robot approaches food from above and feeds someone behind it (sitting in the wheelchair), it no longer needs to be constrained to right-to-left motion. As described in Section 3, the definition of the food frame is ambiguous with respect to a $\pi$ rotation about the axis of gravity. Since the food location was fixed, we manually broke this symmetry by choosing the orientation that was easiest for the on-board planning algorithms.

---

[2] https://github.com/personalrobotics/corl23_towards_general_food_acquisition

Further, some in-food grasp motions that humans executed, specifically tiny rotations of the fork ($3°$ or less), produced negligible motion of the robot with significant planning and collision checking time; so we truncated those rotations to $0°$.

# 5 Experiment 1: Action Evaluation

This experiment was designed to test the utility of the discrete 11-action space on a variety of food items. Previous work in assisted feeding suggests that, depending on level of mobility impairment, users can generally tolerate up to 20% failure rate in food acquisition [5]. Our hypothesis was two-fold: (1) *Coverage:* for each food item, at least one action would meet or outperform baseline performance and meet the 80% user threshold. (2) *Minimal Bad Actions:* each action would have acceptable performance on at least one type of food. If the action set lacks coverage, it is likely too small to adequately acquire new food items in the home, while if there are many bad actions, it is likely too big and will be difficult to use for online learning.

Our hardware setup is summarized in Fig. 2b, and details, including the hardware description, trial description, and success metric are fleshed out in Appendix C.

## 5.1 Experiment Design

We evaluated our action space on 14 diverse food items. Some food items were identical to those used during the human acquisition data collection (Section 4): fries, broccoli, mashed potatoes, spinach mix, and jello (cut into $\sim 1.5$cm slices to obviate the need for cutting). Some food items had similar properties to those in the human data collection with different visual characteristics: powdered doughnut holes, white rice, white bread sandwich, and flat noodles. Finally, some food items were new: baby carrots, grapes, half-strawberries, banana slices, and kiwi slices.

The baseline action set consisted of 3 skewering techniques pulled from past fork-acquisition work [7]. *Vertical skewer* (VS) orients the handle of the fork to be orthogonal to the table and moves straight down applying up to 15N of force before moving straight back up. *Tines vertical* (TV) orients the tines of the fork to be orthogonal to the table and again applies 15N of force straight down before moving back up. Finally, *tilted angle* (TA) orients the handle of the fork 45 degrees off the table normal with the fork flat facing up and approaches the food at that same angle, moving straight upwards after skewering.

For each food item, we perform 10 trials with each of the 3 baseline and 11 human-informed actions for a total of 14 actions $\times$ 14 food types $\times$ 10 = 1960 trials. At about 1 minute per trial, data collection took about 33 hours.

## 5.2 Results

These results are summarized in Fig. 3(Left). All error bars represent Wilson Binomial Proportion 95% Confidence Intervals ($n = 140$ in aggregate, $n = 10$ per food item). Overall, the best action for each food item from the human-informed set significantly outperforms both the best action from the baseline set and the the user-defined benchmark with a success rate of 94.6% ($p < 0.05$ necessarily by non-overlapping confidence intervals).

**Coverage**   All food items except for spinach exhibited a success rate of 90% or higher within the new action space, exceeding the 80% user benchmark. Single-leaf spinach, difficult to acquire due to its thinness, came close with a 70% success rate with the best human-informed action. The nearly complete coverage suggests that this action space is large enough to handle the variety of food items necessary for in-home deployment. Additionally, reducing $k$ maintained a subset of the $k = 11$ medioids down to at least $k = 5$, and so coverage is achieved for $k \geq 8$, as below that, the action space does not include the only action that covers jello.

**Bad Actions** Almost all human-informed actions exhibited good performance on at least one food item. Actions 0, 1, 2, 3, 6, 8, and 10 were the optimal action for spinach, carrot, banana, strawberry, potato, jello, and sandwich respectively. While not an optimal action for any food item, actions 4, 5, and 7 exhibited $\geq 70\%$ success on at least one food item. The only exception was action 9. This action captured the "cutting" motion that humans used on the full, undivided jello cups. Therefore, only the side of the fork comes into contact with the food, making success less likely. That 10/11 actions exhibited good performance suggests that this action space is not excessively large.

**Baseline Comparison** Carrots, grapes, strawberries, bananas, and broccoli exhibited good ($\geq 80\%$) performance with the optimal baseline actions that were designed for them in previous work [4], with the human-informed actions performing as well or slightly better. Sandwich, fries, noodles, and rice exhibited $60 - 70\%$ baseline performance, with insufficient in-food contact (e.g. not enough of the fork present inside of every sandwich layer) as the primary failure mode. These failures were remedied by the increased in-food grasp motion of the human-informed actions. Finally, jello and spinach were completely impossible for the baseline actions to acquire. Jello needed a significant rotation during extraction to prevent the heavy chunk from slipping off the fork. Spinach needed a significant lateral force during the grasp phase to wedge the fork between the flat leaf and the plate. Finally, the human-informed actions were generally able to acquire a greater mass of rice (258mg vs. 212mg) and potato (890mg vs 5930mg) than TA, the only baseline action with any form of scooping-like motion.

## 6 Experiment 2: Online Action Selection

As described in Section 1, we assert that online learning is a necessary component of any in-home robot food acquisition system to handle previously-unseen food items. This is supported by the extensive in-lab evaluation time required for the previous experiment. In this experiment, we evaluate the ability of an off-the-shelf online learning procedure to identify sufficiently good human-informed actions for acquiring previously unseen food items. Our hypothesis is that, given that there are only 11 actions (and as few as 4 can cover the space), such a system should be able to reach the user benchmark on the order of 11 trials for each new type of food. At about 1 minute per trial, in an assistive feeding context, this would happen well within the bounds of a 20-30min meal.

### 6.1 Learning System

As in Experiment 1, this experiment constituted a series of trials. We cycled through all 14 food items, with 1 trial per item. A group of 14 trials, one per food item, constitutes a *round*. Data collection ceased once the success rate over the course of a round exceeded 90% (approaching the 94% optimal). For each trial, food orientation, manual perception, action execution, and success definition, are identical to what is described in Section 5.1.

Our online learning procedure models food acquisition as a contextual bandit [44, 23] with visual context and augmented with haptic post hoc context based on related work [8]. Specifically, each trial prior to action selection, we manually annotate a bounding box around the food item in the RGBD image. The cropped image is run through the SPANet [7] model to create a feature vector that constitutes the visual context. During the approach phase of action execution, we collect raw 6D force-torque data, isolate the period immediately following food contact using a z-force threshold, and run the result through a HapticNet multi-layer perceptron model [6] to generate a feature vector that constitutes the haptic context.

We assume a linear relationship holds between both forms of context and the expected reward (i.e. the success rate) of each of the 11 actions. Therefore, we utilize LinUCB [45], which uses the data collected so far to assign an upper confidence bound to the reward of each action and optimistically selects the action that has the highest upper confidence bound. Previous work [8] suggests that, despite being collected after action selection, optimizing the linear model with the haptic context can decrease the time needed to converge to the optimal action.

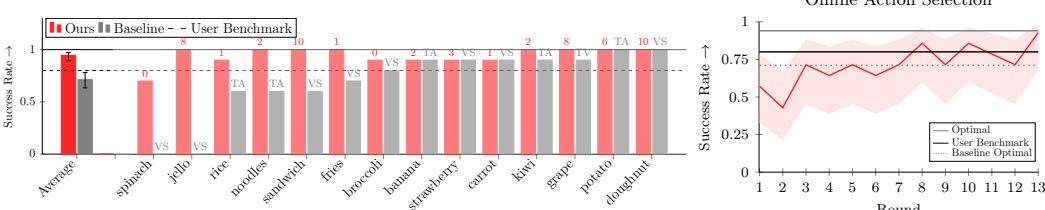

Figure 3: *(Left)* Best action for each food item from both the baseline and our action spaces. The specific action is labeled above each bar. *(Right)* Acquisition success rate for each round of 14 trials (1 per food item) using LinUCB. Error bars represent the 95% confidence interval.

## 6.2 Results

These results are summarized in Fig. 3(Right), where we plot the success rate across all 14 food items in each round (with Wilson 95% Confidence Interval, $n = 14$). Since most human-informed actions perform well on most food items, we see that we get a 50% success rate even in the first round. By round 8, the action performance is on par with the user benchmark. And by round 13, we have approached the expected optimal performance with this action space (i.e. the Average shown in Fig. 3). At one minute per trial, this suggests that this system can successfully learn an acceptable acquisition action for 2-4 new types food within the span of at 30min meal, assuming that all foods of a given type have a similar success rate for each action. And that number is likely higher for foods with similar haptic properties. Most food items converged to a sufficiently good action (e.g. Action 1) with 5 rounds. The food items that took the longest to learn were the haptically "unusual" food items like jello, rice, noodles, and mashed potato, which exhibited particularly poor performance on the skewering actions that worked well on the firmer food items.

## 7 Limitations and Discussion

In this work, we present a methodology to use human trajectory data to identify a subset of food acquisition actions that can acquire a wide variety of food items for robot-assisted feeding applications. The 11 actions we distill from our publicly available dataset [43] are sufficient to pick up 14 food items including hard carrots, soft bananas, slippery jello, compound sandwiches, and continuous mashed potatoes. And the set is so small that we can reasonably expect to determine the optimal action for 2-4 food items over the course of a 30-minute meal.

A major avenue for future work is evaluation in a real in-home context. The foods selected for both the human data collection and on-robot experiments were motivated by surveying a participant and co-designer with mobility impairments about their eating habits. We believe they cover a wide variety of rheological contexts, but there may still exist food types that are sufficiently distinct as to not be covered by the actions presented here. Future work in the home can help identify such foods. These can possibly be addressed with online action space expansion. For example, the caregiver can provide some kinesthetic demonstrations that can be mapped into the action schema and averaged. Additional work can investigate the coupling between these actions and the bite transfer process [4].

One significant hurdle to any future in-home experiments is food perception, a prerequisite for any long-term in-home deployment. In this work, food localization was done manually by clicking on the food center. Autonomous food perception is currently being investigated utilizing general segmentation models [46] and could introduce errors that lower the overall success rate.

In general, the interaction between sources of error (e.g. perception, calibration, handling arbitrary food orientation, environmental hazards like obstacles or a non-level table) and optimal action selection is a key realm for future investigation. Since many foods were well-covered by multiple actions, it is possible that the current action space will be robust to such disturbances, but further study is needed. Overall, this work represents a step towards in-home general food manipulation in both feeding and food preparation contexts, and we hope that the provided method, data, and actions can help enable in-home experimentation in the near future.

**Acknowledgments**

This work was (partially) funded by the NSF GRFP (DGE-1762114), National Science Foundation NRI (#2132848) and CHS (#2007011), the Office of Naval Research (#N00014-17-1-2617-P00004 and #2022-016-01 UW), and Amazon.

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

## A    Human Data Collection

We ran a user study to identify which actions within this 26 dimensional schema are effective for acquiring a diverse food items.

### A.1    Study Design

The study involved participants acquiring bite-sized pieces of a variety of food items with a fork and feeding them to an actuated mouth. The choice of food items was informed by ongoing collaboration with an end-user with C1 quadripledia[3], who had his caregiver take pictures of all the meals he ate in a week. One researcher then grouped similar food items (e.g., bread bun and bagel), resulting in a final set of 13 diverse food items: bagel chunks, mini sub sandwiches, pizza, chicken tenders, fries, broccoli, glazed doughnut holes, mashed potatoes, lettuce, spinach mix, whole jello, instant ramen noodles, and brown rice with beans. The bagels, sub sandwich, and pizza were pre-cut into bite-sized chunks, building off of past research that found that users are okay with caregivers cutting their food into bites before the robot feeds them [47]. The same brand ingredients and preparation procedure were followed for every food item.

The study space consisted of a table with a plate of food on it, a fork near the plate, a chair for the participant to sit in, and an actuated mouth to the left of the chair[4]. An RGB-D camera[5] above the plate captured visual aspects of the food. The fork, table, RGB-D camera, and actuated mouth all had motion capture markers on them, which were tracked by a motion capture system[6] in front of the table. The fork was also actuated with a force-torque sensor[7] to measure haptic aspects of the participant's acquisition action. An experimenter sitting behind the motion capture system, in full view of the participant, oversaw data collection. Fig. 2a shows the study setup.

When each participant arrived, they were first briefed on the study and given time to read and fill out a consent form. They were then given a chance to familiarize themselves with the fork[8] by feeding baby carrots to the actuated mouth. When participants were ready, the actual data collection began, where they were provided a plate with one of the 13 food items, in randomized order, and were asked to feed one bite at a time to the actuated mouth. For each bite, participants were first asked to hold the fork in a comfortable "ready" position above the plate. When the experimented said "start," they lowered the fork to acquire the food item, moved it to the actuated mouth, and held it

---

[3]C1 quadriplegia refers to paralysis of all four limbs as a result of an injury to the first, or top-most, cervical vertebrate.

[4]Because all participants happened to be right-handed, this positioning allowed them easy access to feed the actuated mouth.

[5]Intel RealSense D415

[6]OptiTrak V120 Trio

[7]ATI Industrial Automation 6DOF Nano25 with Net F/T Interface

[8]Due to the force-torque sensor and motion capture dots, the fork was a different shape and weight from regular forks

there until the experimenter said "stop." For each plate of food, participants were asked to feed at least 5 bites, and possibly more if the motion capture system lost tracking of the fork during the bite. In total, each study session took one hour and participants were compensated with a $10 gift card. Three researchers ran the study, and the study procedure was approved by our university's IRB. We had 9 participants, who all happened to be right-handed.

## A.2   Dataset and Qualitative Observations

For each bite the participant acquired, our time series data consisted of the fork pose, force-torque sensor readings, and RGB-D images of the plate. We first cleaned this data by removing mistrials (i.e., trials with missing or corrupted data) and transforming all poses to a uniform frame of reference. We then published this dataset [43] to facilitate future research in food acquisition strategies. This dataset consists of 496 trials, totaling over 1.25 hours of food acquisition data across 13 food items and 9 participants.

Similarly to previous work [6], we observed some patterns in user interaction during data collection.

- Different participants held the fork differently (thumb in front of or behind the fork), which gave rise to different acquisition actions;

- Some food items could be acquired with multiple types of actions, e.g., some users scooped noodles whereas others twirled them;

- Users often tilted the fork while putting downwards pressure on it, in order to get it to pierce the food (e.g., broccoli).

Some of these lead to emergent behaviors identified in Section 4.2.2

## B   Action Schema Point Extraction from Human Data

**Exclusion Criteria**   We excluded any trial where tracking of the fork tip was lost for more than $0.5$ seconds, or where the motion capture system did not read the stationary object poses (e.g., table, mouth) for the entire trial. After exclusions, we had a total of $410$ trials.

**Pre-processing**   To remove noise from the motion capture system, we smoothed all the fork tip poses by applying a median filter of $0.33$ seconds separately to the x, y, z, roll, pitch, and yaw of the pose.

**Significant Timestamps**   We first extracted the significant timestamps of the user's acquisition action. Specifically, we defined the *contact time* as the first timestamp when the distance from the fork tip to the camera exceeded the distance from the pixel corresponding to the fork tip to the camera in the initial depth image of the plate (i.e., the first time the fork pierced the surface of food on the plate). We then worked backwards from contact time, defining *start time* as the end of the $0.5$ sec interval where the fork tip was consistently within a sphere of radius 5 cm, was more than 5 cm from its lowest point, and was more then 35 cm from the mouth. This criteria was based on our experimental design, where we asked the participant hold the fork stationary at a "ready" position before they acquired the food. In the event of multiple such periods where the fork was held stationary, we chose the one where the fork was the highest. We then defined *end time* to be the last timestamp when the fork was within 7 cm from its lowest point. And finally, we worked backwards from end time, defining *extraction time* to be the latest time when the fork was 1 cm away from its lowest point, within 2 sec of the end time.

All-in-all, the motion the participant took between *start time* and *contact time* corresponds to their **pre-grasp** motion, the motion they took between *contact time* and *extraction time* corresponds to their **grasp** motion, and the motion they took between *extraction time* and *end time* corresponds to their **extraction** motion.

**Food Reference Frame** Although a robot will know the food it is targeting before beginning its motion, with human data we have to extract the food item they were targeting from the data. We did so by segmenting the visually separate food items in the first RGB image of the plate of food. First, we detected the plate by: (a) using inpainting to remove glare; (b) using k-means clustering (k=3) to simplify the colors; and (c) finding the largest contour of the image that has at least $50\%$ blue pixels. In practice, this reliably detected the plate for every food item in our study. We then masked out all the non-plate pixels from the de-glared image, and detected the food bounding box by: (a) running k-means (with k=2 for most food items, and k=3 for broccoli since its colors were closest to blue) to simplify colors; (b) masking out all the blue colors; (c) narrowing the mask to separate touching food items; (d) computing contours; and (e) fitting rotated rectangles to every contour with an area between within a hardcoded range. In practice, this reliably segmented separate food items, like bagel pieces or chicken tenders. For food items with a lot of overlap, like fries, this approach sometimes segmented multiple pieces of fries as the same. However, since participants often also acquired multiple overlapping pieces of those food items, we accepted those slight errors in food detection. For foods that weren't separated into bites like noodles or mashed potatoes, this algorithm rightly segmented it as one contiguous chunk of food.

Once we segmented separate bites of food, we defined the food reference frame to be centered at the center of the bounding box the fork tip was in at contact time, rotated to align with its major axis (i.e the center of the bite the user selected.)

**Pre-Grasp** The above preliminaries enable straightforward extraction of the pre-grasp, grasp, and extraction components of the action schema. For pre-grasp, we computed the target offset as the the target offset as the fork position at contact time in the food reference frame. We computed the initial utensil transform by taking the fork's linear velocity during a $0.5$ second window before contact time and extrapolating that backwards $0.1$ m, with a fixed orientation. And we took the force threshold to be $50\%$ of the max force between start time and contact time.

**Grasp** We defined the in-food twist to be the transformation between the fork pose at extraction time and contact time, and the duration of the twist to be the duration between extraction time and contact time. We defined the force and torque thresholds to be $50\%$ of the max force and torque between contact time and extraction time.

**Extraction** We defined the out-of-food twist to be the transformation between the fork pose at end time and extraction time, and the duration of the twist to be the duration between end time and extraction time.

**Cleaning** Three trials resulted in values of NaN or inf for at least one of the dimensions of the action schema. We eliminated these, resulting in $407$ actions used for clustering.

## C  Experiment Setup Details

Our experimental setup is shown in Fig. 2b and was performed with a 6 DoF JACO2 robotic arm [48] with a 3D-printed handle and fork-shaped end-effector. To implement the force/torque thresholding, we instrumented the fork with a 6-axis ATI Nano25 Force-Torque sensor [49]. The center of the food (and the bounding box used for visual context in the online learning experiment described in Section 6) were annotated manually from the robot's eye-in-hand vision system. This was done in order to run a controlled experiment specifically focused on food acquisition, as opposed to introducing additional variance with a (possibly imperfect) food perception system. This system includes the Intel RealSense D415 RGBD camera and the NVidia Jetson Nano for wireless image transmission. Food was placed on a plate equipped with an AprilTag [50] for camera calibration and mounted on an anti-slip mat commonly found in assisted living facilities [51].

In each trial, the food placed in the vicinity of the AprilTag on the plate and oriented such that the major axis of the bounding ellipse is parallel to the bottom edge of the fiducial. The end-effector

moved to a fixed position above the plate, and the location of the center of the food is annotated manually in the RGBD camera image. After action execution, we wait at least 3s before recording success or failure. For most food items, success is defined as the entire item being removed from the plate. If a homogeneous food item breaks (a common occurrence with banana slices), at least half of the item needs to end up on the fork. For the sandwich, success required that all layers (both pieces of bread, the lettuce, and the cheese) make it off the plate. Finally, for multi-piece and continuous items (i.e. potatoes, rice, noodles), a conservative success metric was set at 200mg (∼15 grains of rice, or 1 full noodle).

