# OpenReview forum: "Towards General Single-Utensil Food Acquisition with Human-Informed Actions"
_robot-learning.org/CoRL/2023/Conference — CoRL 2023 Poster_

### Official Review · Reviewer_W1Qt · 2023-07-16

**Confidence:** 4
**Originality:** Very Good
**Technical Quality:** Very Good
**Clarity Of Presentation:** Excellent
**Impact:** 3

**Recommendation:**

Weak Accept: I recommend accepting the paper, but will not argue for my recommendation if the majority of other reviewers have a different opinion.

**Review:**

Strengths:
- The method is very performant compared to prior art and seems tractable to scale to novel food assisted feeding scenarios
- The overall integrated system is impressive and much closer to deployable real world impact compared to other CoRL works.
- This work incorporates many practical motivations from realistic use cases, such as sampling training/test objects from assisted-feeding patients, extracting actions from real human feeding as opposed to robot teleoperation, and using data from in-home robot-assisted feeding user studies. This will make the work more directly applicable and impactful for practitioners.
- The presentation is polished and very easy to understand the core contributions

Weaknesses:
- The most significant contribution is the action extraction method, but there were insufficient details in Section 4.2.2. More analysis would be helpful, such as the choice of K=11 learned actions: is the method sensitive to clustering algorithm? Would far more or far fewer learned actions affect downstream learning? Can all 410 extracted actions be used during online learning instead of 11 extracted medoids?
- The contextual bandit setting seems overly restrictive, but I will note that my understanding of the food acquisition field is limited and I do note that the prior work [6] in the paper also make this assumption. However, in cases such as prior works that probe food as part of the action representation, it seems like the proposed framework may not be able to support such types of interactive action primitives.
- The reliance on user-provided bounding boxes as well as RGBD are limitations not found in other types of learning based robot manipulation systems.
- While this paper is extremely relevant and impactful for robot-assisted food acquisition, it's not clear how the technical contributions contained could extend to more general domains (this is not really a hard criticism, since the authors acknowledged this up front).

**Quality Of The Limitations Section:**

Limitations are addressed clearly

**Questions For Rebuttal:**

My main questions could be clarifications regarding the weaknesses above. In addition, some situation of this work with skill learning robotics papers would be helpful.

**Robotics Focus:**

Sufficient demonstration on hardware

**Summary Of Paper:**

This work introduces a method for extracting a set of food acquisition primitives from human demonstration data, and shows that such a set of food acquisition actions is sufficient for acquiring a large variety of realistic real world food items. In a setting where it is assumed that there exists a single action primitive that is sufficient and optimal for acquiring a target food item, this work aims to learn the set of possible action primitives. The contributions have three components: 1) an over-parameterized 26-D action space for capturing human food acquisition techniques with three stages: approach, grasp, and extraction, 2) a 1.25 hours of food acquisition dataset from human participants, 3) a method for extracting a smaller number of food acquisition primitives from such a dataset. This work compares 11 extracted actions from the human dataset with prior works that use 3 primitives. The data-driven action space can perform well and even generalize to unseen object types.

**Summary Of Recommendation:**

This paper is well motivated, technically sound, and introduces non-trivial advances compared to prior works in robot-assisted feeding. Namely, expanding the flexibility and diversity of action primitives is clearly motivated from advances in other parts of robotics, and the empirical results in this work are strong. I particularly am impressed by the real world motivation with user studies in various aspects of this work, resulting in a work that will be significant within the robot-assisted feeding domain. In addition, the presentation is polished. My main considerations would be to explain the technical contribution a bit more.

[Post-Rebuttal Update]
The reviewers have addressed my questions and I maintain my positive rating of Weak Accept.

---

### Official Review · Reviewer_tu76 · 2023-07-19

**Confidence:** 4
**Originality:** Good
**Technical Quality:** Good
**Clarity Of Presentation:** Very Good
**Impact:** 3

**Recommendation:**

Weak Accept: I recommend accepting the paper, but will not argue for my recommendation if the majority of other reviewers have a different opinion.

**Review:**

Strengths:
- The paper is very well written and the experimental evaluation is quite thorough. I appreciate the number of food items tested on, the diversity of physical properties, and the separation of experiments into success rates on seen food items and unseen evaluation with the bandits approach.
- A huge problem in robot-assisted feeding is expert design of primitives which can handle a range of food items, and this paper proposes a very natural way to avoid doing so while leveraging demonstration data effectively.
- The design of the continuous action space was thoughtful and I think provided a happy medium of covering a range of parameters without exploding the action space dimensionality.
- The use of online bandits to infer the optimal action is a clean framework for generalizing to unseen items.
- The supplemental video is helpful for understanding the various actions


Weaknesses:
- My primary concern with this work is that the paper makes strong claims about "general food acquisition" throughout, though the approach is largely geared towards and only tested on skewering -- I would argue that general food acquisition constitutes a range of utensils (spoon, fork, chopsticks, knife, etc.). The action space is largely tailored to skewering specifically (approach, grasp, extract) and cannot easily accommodate other non-quasistatic actions like cutting, twirling, scraping, wiggling, etc. which are all under the umbrella of general food acquisition. The 'cutting' behavior referenced with the jello, for example, is merely a linear movement in the grasping phase and does not generalize to something like cutting steak which would require much higher frequency movement. For many of the items tested (i.e. sandwich, donut, rice, mashed potatoes, noodles), a case also needs to be made for skewering, as it seems that those items would benefit from a different tool/acquisition approach altogether. I suggest reframing the motivation to focus on skewering specifically.
- As the authors pointed out, the food perception in this work was largely simplified to a clean workspace with no distractors and classical/manual techniques to isolate the item of interest. SPANet and other works in the acquisition domain have evaluated on varied degrees of clutter and with more challenging scenarios like the food item being near a plate wall or stacked on top of other items (which would be an interesting testbed for this approach) -- this should be justified.


**Quality Of The Limitations Section:**

Limitations are addressed clearly

**Questions For Rebuttal:**

Can you provide more insight into how 11 clusters were chosen specifically? It would be interesting to see an ablation on the success rate  as a function of k. Additionally, it would be great to better understand the rationale of the 26 continuous parameters. i.e., what happens if the force threshold is omitted, etc.

How many online interactions are required per-food item to infer the appropriate action? And is the food item replaced between interactions (otherwise, could something delicate like a banana slice keep getting damaged over time -- and the optimal action may also change?)

Right now, the action clusters are a bit opaque in terms of interpretability -- could you provide some summary of the emergent behaviors in each action?

**Robotics Focus:**

Sufficient demonstration on hardware

**Summary Of Paper:**

This work proposes a novel skewering action space in bite acquisition for robot-assisted feeding. The approach partitions the skewering problem into approach, grasp, and extraction phases, and performs clustering on a dataset of skewering trajectories collected by humans to infer the continous parameters of these motions. In doing so, the dimensionality of the action space is greatly reduced to just 11 discrete actions. This manageable action space provides a small search space such that at test time, the optimal action for an unseen food item can be inferred in a bandits style approach within a small number of trials.

This approach is tested on a variety of food items ranging in visual and haptic properties, and the proposed approach achieves around 80-90% success on the difficult task, outperforming prior work which relies on discretized primitives instead.

**Summary Of Recommendation:**

Overall, the technical contribution is good and original compared to prior work. The paper is written very well, and the experimental evaluation is quite thorough. However, the paper should avoid making strong claims about the contributions and instead focus on the skewering subproblem, while providing more insight into the design choices in terms of the action space dimensionality.

---

### Official Review · Reviewer_tE7u · 2023-07-20

**Confidence:** 2
**Originality:** Good
**Technical Quality:** Good
**Clarity Of Presentation:** Very Good
**Impact:** 3

**Recommendation:**

Weak Accept: I recommend accepting the paper, but will not argue for my recommendation if the majority of other reviewers have a different opinion.

**Review:**

# Strengths
- This work presents an interesting insight: the space of the 26-dim action schema, when perceived like a skill space, can be reduced to just 11 meaningful skills for the tasks considered. Broadly, this suggests that for many practical tasks, a few discrete number of strategies are enough to solve them.
- I appreciate the care in choosing the food items based on the meals an end-user with C1 quadripledia. This sufficiently justifies the seemingly restricted variety of food items considered (14 test items) in this work, as the benchmark's range is grounded to this real world application.
- The results show that the use of human data to extract useful actions (or skills) is better than pre-defining the actions for this task. This justifies the role of learning from data in achieving robustness in real-world tasks as compared to pre-defined primitives.
- The videos and extracted actions provided in the supplementary are informative and appreciated.


# Weaknesses
- While the insight is interesting that only a few distinct strategies are enough to solve the task of picking up food objects using forks, this could very well be a commentary on the difficulty of the particular food acquisition task itself. Naturally, as the task gets more complex, the 11 selected schema would not be enough. In a way, this paper shows that the baseline composed of 3 strategies was not enough for the task considered and one indeed needs 10-11 strategies to pick up the food items considered. As the paper suggests in limitations, as sources of error (perception, food orientation) and food varieties increase, even for the limited task definition considered here, one would expect more actions are needed to solve the task. Therefore, it is hard to take much away from the paper's key insight, other than that "simple tasks require fewer distinct strategies to solve and harder tasks require more" — which is not as surprising as the paper originally projects.
- Baseline actions: Since the baseline actions also perform quite reasonably on the task, even without any human data, is it possible that if one or two manually engineered baseline policies are added, then the performance is matched with this paper's results? For exanple, since jello and spinach are the challenging tasks due to rotation or wedging, then manually modifying the action schema to incorporate these new actions would be a more efficient way to get to a working set of actions, than to collect human data. A contextual bandit can be trained on such an action space too, and thus generalize to new food.
- Formulation as contextual bandit: Since the task of fork lifting of food is a relatively static and simple task, it can be formulated as a contextual bandit for one time action selection. But ideally, an agent should be reactive to changes in the environment dynamics. Therefore, a reactive policy is necessary to make food acquisiton more robust and generally applicable. However, for RL, it is unclear how the "action" or parameterized-skill space extracted in this work from user data, can be used.
- The action schema is heavily engineered for this task and convert one trajectory into one action / skill with 26 parameters. Even the food localization in pixel space is done manually around the food center, which provides a lot of engineered information to the learning modules of this work. Would the insights of this work still be relevant when the tasks are harder or long horizon, and the action schema is not engineered?

**Quality Of The Limitations Section:**

Additional details required

**Questions For Rebuttal:**

- Since I am not knowledgeable in the domain of food acquisition application, I cannot fully appreciate the complexity of this task. What are the learning problems that make this task hard? Looking at the videos, it seems that main challenge is to generalize over the objects in the image space (given as state input). What are the other challenges?
- It is not clear to me what is the role of the HapticNet feature vector in predicting the action to take. If the arm is already in the approach phase, this means one of the 11 actions has been selected and executed. And since this is just one-step task, how and when is the HapticNet feature vector utiized?
- What is the result on the unseen food categories for the proposed method and the baselines? Does the agent primary lose performance on the unseen food items?

**Robotics Focus:**

Sufficient demonstration on hardware

**Summary Of Paper:**

This work centers on food acquisition as a task and posit that a small number of action strategies (11 parameterizations of a pre-defined action schema, including planning capabilities) are sufficient to acquire a variety (14) of food items with >=80% success rate. These 11 action strategies are identified by collecting human user food acuquisition data, mapping them to the action schema (26-dimensional) and clustering the space of actions. Finally, by treating the task as a contextual bandit (state = feature vector of food plate image + haptics), they show the optimal action out of the 11 actions can be found quickly ~in 20-30 minutes for 2-4 food items.

**Summary Of Recommendation:**

While I don't believe there are many generic robot learning insights that can be derived from this work, I do appreciate the experimental rigor, choice of application, and demonstration of results on a real robot and real food. The videos seemed simple at a glance, but robust and reliable execution is important to be shown. I am unsure about the applicability of CoRL as the right conference for this application-engineered paper, but I don't want to be the gatekeeper of simple but potentially impactful empirical work, and thus would recommend weak acceptance, contingent on peer reviewers and AC agreeing to the importance of the empirical contributions of this work.

---

### Official Review · Reviewer_kQNo · 2023-07-20

**Confidence:** 4
**Originality:** Good
**Technical Quality:** Very Good
**Clarity Of Presentation:** Very Good
**Impact:** 3

**Recommendation:**

Weak Accept: I recommend accepting the paper, but will not argue for my recommendation if the majority of other reviewers have a different opinion.

**Review:**

Pros:
1. Paper is well written, technique is nicely explained and details and procedures are clear.
2. The results are also correct explained and seemed intuitive to understand.

Con:
1. There's a lot of assumptions in the system design, which works very well for the constrained setting, but its unclear how this can scale.
2. Both experiments and results are highly domain specific, so my one suggestion to improve the paper could be, how would you inform robotics in general based on the observations of this study

**Quality Of The Limitations Section:**

Additional details required

**Questions For Rebuttal:**

1. What could be a way to scale this?
2. How would you inform robotics in general based on the observations of this study?
3. Given that the title mentions human feedback, is there any user interaction observations from this study?

**Robotics Focus:**

Sufficient demonstration on hardware

**Summary Of Paper:**

The method proposes various ways of picking up food from a plate using a robotic end effector using online learning policies. This is an effort that can serve with people with disabilities, I love that robotics is being used for good. They propose that a small number of actions can pick up almost any type of food. This seems consistent with my personal experience. As such, it is a data constrained regime(only 33 hours). The frame definitions are defined with respect to the plate rather than the end effector, that appeared surprising.

**Summary Of Recommendation:**

This paper studies the problem of food acquisition and learning of that manipulation from human actions. The paper is technically sound and well executed, also well explained. This is why the accept recommendation. There is also hardware experiments. However, the study is within a very constrained setting. The system makes assumptions that may not generalize to all homes, but it still shows incredible generalization to many different food types. There are also many hand engineered aspects that I feel will not scale, hence the weak accept.

---

### Decision · Program_Chairs · 2023-08-30

**Decision:**

Accept (Poster)

**Comment:**

The reviewers agree that this paper brings an insightful and worthwhile contribution to the CoRL community. The paper is well-written, the experiments tests a large number of food items, and insights around lower-dimensional action spaces are appreciated. The reviewers also brought up several points of feedback that the authors are encouraged to address, including but not limited to: (a) changing the title and the rest of the writing to scope the contribution to skewering (with a fork) rather than general food acquisition, (b) discussing the limitations around the system assumptions/task-specific engineering, and (c) including more discussion of the general takeaways of the paper.